# STEAP1–4 (Six-Transmembrane Epithelial Antigen of the Prostate 1–4) and Their Clinical Implications for Prostate Cancer

**DOI:** 10.3390/cancers14164034

**Published:** 2022-08-20

**Authors:** Michael Xu, Latese Evans, Candice L. Bizzaro, Fabio Quaglia, Cecilia E. Verrillo, Li Li, Julia Stieglmaier, Matthew J. Schiewer, Lucia R. Languino, William K. Kelly

**Affiliations:** 1Sidney Kimmel Cancer Center, Thomas Jefferson University, Philadelphia, PA 19107, USA; 2Department of Pharmacology, Physiology and Cancer Biology, Thomas Jefferson University, Philadelphia, PA 19107, USA; 3Department of Urology, Thomas Jefferson University, Philadelphia, PA 19107, USA; 4Department of Pathology, Thomas Jefferson University, Philadelphia, PA 19107, USA; 5Early Development Hematology/Oncology, Amgen Research (Munich) GmbH, 81477 Munich, Germany; 6Department of Medical Oncology, Thomas Jefferson University, Philadelphia, PA 19107, USA

**Keywords:** prostate cancer, six-transmembrane epithelial antigen of the prostate, biomarker, immunotherapy, cancer vaccine, T-cell-engaging antibody

## Abstract

**Simple Summary:**

Despite recent therapeutic advances in the treatment of prostate cancer, metastatic castration-resistant prostate cancer continues to cause significant morbidity and mortality. New research into highly expressed proteins in metastatic castration-resistant prostate cancer shows that Six-Transmembrane Epithelial Antigen of the Prostate 1–4 (STEAP1–4) are significant drivers of prostate cancer aggressiveness and metastasis. STEAP1, in particular, is highly expressed on the plasma membrane of prostate cancer cells and has received significant attention as a potential therapeutic target. This review highlights what is known about STEAP1–4 and identifies knowledge gaps that require further research.

**Abstract:**

Six-Transmembrane Epithelial Antigen of the Prostate 1–4 (STEAP1–4) compose a family of metalloproteinases involved in iron and copper homeostasis and other cellular processes. Thus far, five homologs are known: STEAP1, STEAP1B, STEAP2, STEAP3, and STEAP4. In prostate cancer, STEAP1, STEAP2, and STEAP4 are overexpressed, while STEAP3 expression is downregulated. Although the metalloreductase activities of STEAP1–4 are well documented, their other biological functions are not. Furthermore, the properties and expression levels of STEAP heterotrimers, homotrimers, heterodimers, and homodimers are not well understood. Nevertheless, studies over the last few decades have provided sufficient impetus to investigate STEAP1–4 as potential biomarkers and therapeutic targets for prostate cancer. In particular, STEAP1 is the target of many emerging immunotherapies. Herein, we give an overview of the structure, physiology, and pathophysiology of STEAP1–4 to provide context for past and current efforts to translate STEAP1–4 into the clinic.

## 1. Introduction

Among males worldwide, prostate cancer (PCa) is the most commonly diagnosed malignancy and second leading cause of cancer-related death [1]. While the majority of PCa patients can be cured with local therapy, some patients progress with rising prostate-specific antigen (PSA) values and develop metastatic disease [2]. Androgen ablation (e.g., androgen deprivation therapy) is the cornerstone treatment for metastatic hormone-sensitive PCa. However, a majority of these patients will relapse and develop metastatic castration-resistant prostate cancer (mCRPC). The changes that underlie the transition of PCa to mCRPC include, but are not limited to, inactivation of tumor suppressors, changes in antigen expression, and androgen receptor (AR) mutations [3]. Because mCRPC is aggressive, it is associated with a poor prognosis.

Given the lethality of mCRPC, much effort has been dedicated to understanding its biology. The medical oncology community has made great strides in developing new treatments for mCRPC, which include novel androgen-directed therapies, chemotherapies, and radiolabeled therapies targeted to cell-surface antigens (e.g., prostate-specific membrane antigen). However, males with mCRPC continue to progress, and there is a critical need to find new biomarkers and targets for PCa.

A unique and promising therapeutic target is Six-Transmembrane Epithelial Antigen of the Prostate (STEAP), a family of metalloproteinases that are variably expressed in normal prostate tissue and, except STEAP3, overexpressed in PCa [4]. Since STEAP1 was first reported by Hubert et al. in 1999 as a cell-surface antigen that is overexpressed in PCa, several homologs (STEAP1B, STEAP2, STEAP3, STEAP4) have been discovered [5,6,7,8,9]. However, their roles in PCa development are unclear. This review provides an overview of the structure, physiology, and pathophysiology of STEAP1, STEAP1B, STEAP2, STEAP3, and STEAP4 (STEAP1–4) in order to provide context for their clinical implications in PCa diagnosis, prognosis, and treatment.

## 2. Structure and Physiology of STEAP1–4

In mammals, iron and copper exhibit several biochemical similarities: Both alternate between two oxidation states, are absorbed in the small intestine via divalent metal transporter 1 (DMT1), and are cofactors for several enzymes (e.g., iron in catalase) [10,11,12]. Iron and copper metabolisms are intricately interconnected, often acting to reduce/oxidize each other (e.g., in cytochrome c oxidase). Importantly, dysregulation of iron and copper homeostasis is associated with cancer progression [13]. Because tumor cells proliferate more rapidly than normal cells, their iron and copper demands exceed those of normal cells. Therefore, there is significant interest in understanding the mechanisms that allow tumor cells to upregulate iron and copper uptake.

STEAP1–4 are metalloproteinases that play important roles in iron and copper homeostasis [4]. The C-terminal domains of STEAP1 and STEAP1B are homologous with *Saccharomyces cerevisiae* ferric reductase (FRE), while the N-terminal and C-terminal domains of STEAP2-4 are homologous with prokaryotic F_420_:NADP^+^ oxidoreductase (FNO) and *Saccharomyces cerevisiae* FRE, respectively [4,8]. In mammals, STEAP1–4 also play roles in regulating cell proliferation and apoptosis, attenuating oxidative stress, and mediating the transferrin cycle [14,15].

The literature is inconsistent concerning the interactions that STEAP1–4 have with each other. STEAP1 can form heterotrimers with STEAP2 and STEAP4; STEAP1, STEAP2, and STEAP4 can form homotrimers; STEAP3 and STEAP4 can form homodimers; STEAP proteins may form heterodimers [16,17,18]. However, the stoichiometries and functions of these heterotrimers, homotrimers, heterodimers, and homodimers are unknown. Perhaps, a reason for the inconsistencies in the literature is that different heterotrimer compositions, e.g., one STEAP1 moiety and two STEAP2 moieties or two STEAP1 moieties and one STEAP2 moiety, result in structural and physiologic properties. Furthermore, different heterotrimers and heterodimers may be variably expressed depending on the tissue/cancer or even the cancer grade. These principles also apply to homotrimeric and homodimeric multimers of STEAP1–4.

In this section, we briefly summarize the structural and physiologic properties of STEAP1–4 (Figure 1). For more information, we refer the reader to a pertinent review [19].

### 2.1. STEAP1 and STEAP1B

STEAP1, which is also called STEAP, is a 339-amino-acid (39.9 kDa) protein with six transmembrane α-helices and intracellular, hydrophilic N- and C-terminal domains [9]. Because STEAP1 lacks the N-terminal NADPH-binding FNO domain found in STEAP2-4, it does not independently exhibit Fe^3+^ or Cu^2+^ reductase activity [4,17,18]. Its *C*-terminal domain contains a *b*-type heme cofactor [16]. Despite its lack of the NADPH-binding FNO domain, STEAP1 curiously adopts a reductase-like conformation. This may be due in part to the homology that STEAP1 shares with STEAP2-4, and perhaps in part to the possibility that STEAP1 may reduce metal–ion complexes and oxygen when it interacts with the NADPH-binding FNO domain of STEAP2 or STEAP4.

STEAP1 is expressed at surface cell–cell junctions [4,9,15]. It is expressed in many tissues, though its expression levels are higher in prostatic secretory epithelium. Although STEAP1 expression was believed to be unresponsive to androgen deprivation or stimulation, Gomes et al. showed that STEAP1 expression is downregulated by 5α-dihydrotestosterone. This downregulation, intriguingly, is AR-dependent [15].

STEAP1B is a truncated homolog of STEAP1. Like STEAP1, STEAP1B does not exhibit Fe^3+^ or Cu^2+^ reductase activity due to its lack of an NADPH-binding FNO domain [8]. It has two associated transcripts, STEAP1B1 and STEAP1B2. STEAP1B1 consists of four transmembrane α-helices and intracellular N- and C-terminal domains, while STEAP1B2 consists of three transmembrane α-helices, an intracellular N-terminal domain, and an extracellular C-terminal domain. Notably, STEAP1B1 mRNA is highly expressed in benign prostate cell lines (e.g., PNT2), while STEAP1B2 mRNA is highly expressed in androgen-dependent PCa cell lines (e.g., LNCaP).

### 2.2. STEAP2-4

STEAP2-4 exhibit Fe^3+^ and Cu^2+^ reductase activity, allowing them to reduce extracellular iron and copper [4,14]. Like STEAP1, STEAP2-4 each consist of six transmembrane α-helices and intracellular, hydrophilic N- and C-terminal domains [20]. The N-terminal contains an intracellular NADPH-binding FNO domain, while the *C*-terminal contains a *b*-type heme-containing FRE domain. The Fe^3+^ or Cu^2+^ reductase activity of STEAP2-4 is mediated by electron transfer from NADPH to membrane-embedded *b*-type heme and FAD^+^, as well as to extracellular metal–ion complexes.

STEAP2, which is also called Six-Transmembrane Protein of Prostate 1 (STAMP1), is a 490-amino-acid (56.1 kDa) protein that is highly expressed in the Golgi complex, trans-Golgi network, and plasma membrane [5,21]. This suggests that STEAP2 is involved in certain endocytic and exocytic trafficking pathways. STEAP2 is also highly expressed in androgen-sensitive, AR-positive PCa, suggesting that it is dependent on AR signaling for expression.

STEAP3, which is also called STAMP3, pHyde, and Tumor-Suppressor-Activated Pathway 6 (TSAP6), is a 488-amino-acid (54.6 kDa) protein that is an essential component of the transferrin–transferrin receptor cycle [6,14,22]. More specifically, STEAP3 is involved in erythroid–transferrin endosome-mediated iron uptake; STEAP3 colocalizes with transferrin, transferrin receptor, and DMT1 [14]. With regards to its other functions, STEAP3 is regulated by a p53-responsive element and interacts with pro-apoptotic factors [6]. These functions suggest that STEAP3 is a tumor suppressor.

STEAP4, which is also called STAMP2 and TNF-α-Induced Adipose-Related Protein (TIARP), is a 459-amino-acid (52.0 kDa) protein that regulates inflammatory responses, fatty-acid metabolism, and glucose metabolism [7,23,24]. Initially identified in a screen of gene products upregulated by TNF-α in differentiating pre-adipocytes, STEAP4 almost completely colocalizes with transferrin and transferrin receptor 1 in the Golgi complex and plasma membrane [4,14]. STEAP4 likely depends on AR signaling for robust expression.

## 3. Pathophysiologic Roles of STEAP1–4 in Prostate Cancer

Although STEAP1–4 are homologs, they exhibit different expression and localization patterns in different tissues. Thus, a complex set of phenotypes is associated with these metalloreductases and, subsequently, a complex set of associated oncogenic properties (Figure 2). The most well-studied STEAP in PCa, STEAP1 is the subject of robust clinical investigation. Herein, we describe the major pathophysiologic roles of STEAP1–4 that are currently known in the literature.

### 3.1. STEAP1 in Prostate Cancer

Relative to normal prostate tissue, STEAP1 is overexpressed in malignant prostate tissue [9]. STEAP1 overexpression promotes prostate tumor growth, and STEAP1 knockdown decreases prostate tumor growth [15,25,26]. No widely accepted mechanism links the properties of STEAP1 with its oncogenic overexpression, but recent articles shed light on some possibilities. STEAP1 seems to act as a channel for small molecules that are involved in intercellular communication [25]. Evidence suggests that STEAP1 acts like connexins (gap junctions). When connexins that are strongly associated with prostate tumor growth (e.g., connexin 43 and connexin 32) are inhibited, intercellular communication shifts to STEAP1-mediated communication [26]. Such communication promotes stromal cell recruitment and proliferation, ostensibly leading to enhanced tumor survival and growth [15]. The role of the *b*-type heme cofactor is unknown, but one can speculate that it reduces molecules that travel through STEAP1.

Since mesenchymal stem cells (MSCs) often express STEAP1, STEAP1 overexpression in PCa may be the sum result of STEAP1 upregulation in malignant prostate cells and MSC-derived PCa stromal cells [26,27,28]. Furthermore, evidence suggests that certain epithelial-to-mesenchymal transition biomarkers are positively associated with PCa aggressiveness [29]. For example, loss of E-cadherin and subsequent loss of cell polarity are associated with higher Gleason scores. Whether STEAP1 overexpression is directly associated with this transition is unknown. STEAP1 overexpression is seen in other cancers as well. For example, STEAP1 overexpression in ovarian cancer ES-2 cells leads to upregulation of the epithelial-to-mesenchymal-transition-related genes (vimentin, N-cadherin, ZEB1, Twist, Snail, Slug) and loss of E-cadherin expression [30]. These changes produce cells with increased invasive, migratory, proliferation, and clonogenic phenotypes.

The implications of these considerations are two-fold. First, STEAP1 overexpression in PCa may be the result of its local tumor microenvironment (rather than solely due to STEAP1 overexpression in primary tumor cells). This may increase the attractiveness of STEAP1 as a biomarker and therapeutic target. Numerous studies, as described later, show that STEAP1 overexpression is immunogenic; enhancing the immune response against STEAP1-overexpressing PCa cells may further induce anti-cancer activity. Second, STEAP1 therapeutics may not be as specific to malignant prostate tissue as previously assumed. There may be off-tumor effects on other tissues that express STEAP1, namely, MSC-derived stromal tissues.

Classically, the Gleason grading system is used to evaluate prognoses in patients with PCa [31]. The Gleason score is equal to the sum of values corresponding to the two most frequent histologic patterns seen in a biopsy sample. In brief, the Gleason score assigned to a histologic pattern increases as the cells become more poorly differentiated. Thus, a higher Gleason score corresponds with a poorer prognosis [32].

Figure 3 gives an example of an immunohistochemical stain of a section of a moderate-grade PCa tumor. It shows that STEAP1 is highly expressed on the prostate cancer cell surface and that immune cells infiltrate in the stroma of PCa tumors. Since STEAP1–4 are highly expressed and localized to PCa tumors, they are a promising avenue for targeted therapy in prostate cancer. Further work is ongoing to better understand the expression and roles of STEAP proteins and immune infiltrative cells in different grades of PCa.

### 3.2. STEAP2-4 in Prostate Cancer

The literature on the association between PCa and STEAP2-4 is scarcer than that for STEAP1. Nonetheless, STEAP2 and STEAP4 are overexpressed in PCa and, thus, deserve attention [33,34,35,36,37].

Overexpression of STEAP2 increases the proliferation of PCa cells, while low expression levels of STEAP2 are associated with decreased PCa cell proliferation [33,34]. STEAP2 seems to promote cell proliferation, at least in part, by upregulating factors (e.g., cyclin H, Ki67) that promote cell-cycle progression and downregulating factors (e.g., cyclin-dependent kinase inhibitor p21) that inhibit cell-cycle progression. This is substantiated by experiments demonstrating that STEAP2 siRNA-transfected cells have a decreased proliferative phenotype. These proliferative and antiapoptotic activities may be linked to STEAP2-mediated upregulation of mitogen-activated protein kinase (MAPK) pathway activity [33]. The expression of various tissue-remodeling proteins, e.g., MMP13, MMP3, SERPINE1, and MMP7, is also affected by changes in STEAP2 expression. Furthermore, STEAP2 overexpression in the normal prostate cell line PNT2 confers a migratory and invasive phenotype, suggesting that STEAP2 overexpression plays a role in driving the metastatic potential of PCa cells [35].

The literature on the association between STEAP3 and PCa is sparse. As of now, the expression and function of STEAP3 are best understood in the context of erythrocytes and erythropoietic tissues [22,38]. However, even then, the evidence is contradictory. For example, data suggest that STEAP3 is a relatively ubiquitous tumor suppressor that induces apoptosis via a caspase-3-dependent pathway [6,39]. However, there are also data that show that STEAP3 promotes iron storage and tumor proliferation in Raji, a Burkitt lymphoma B-cell line, when iron levels are low [40]. In poorly differentiated PCa, STEAP3 is the only STEAP whose expression is downregulated [39]. Interestingly, STEAP3 overexpression in gliomas is associated with increased cell proliferation and tumor-growth phenotype [41].

Overexpression of STEAP4 seems to play a role in PCa progression, as malignant prostate cells abundantly express STEAP4 in the Golgi complex and plasma membrane [7,37]. The function of STEAP4 is not completely understood, but great strides have been made. First, evidence suggests that STEAP4 increases the generation of reactive oxygen species (ROS) via its oxidoreductase activity [36]. The function of these ROS is not completely understood, but they likely induce ATF4 expression. In brief, ATF4 is a transcription factor that is induced by various forms of stress (ER, metabolic, oxidative) and implicated in tumorigenesis [42,43]. Second, STEAP4 promotes the development of mCRPC. In an initial situation where a PCa is castration-sensitive, androgens activate AR. This increases STEAP4 expression, which induces Fe^3+^ reduction and consumes NADPH. The subsequent increase in ROS induces ATF4 expression, which may promote PCa growth and progression, as well as eventual development of castration resistance [36]. Third, STEAP4 is involved in promoting a lipopolysaccharide (LPS)-induced inflammatory microenvironment that promotes PCa proliferation [44]. Although the mechanism underlying this is unclear, evidence suggests that LPS acts via the cyclic guanosine monophosphate–protein kinase G (cGMP-PKG) pathway to increase STEAP4 expression. Fourth, STEAP4 expression in PCa is epigenetically regulated and associated with the inhibition of anchorage-independent cell growth via its association with focal adhesion kinase [45].

STEAP1–4 have many functions outside of or secondary to their intrinsic metalloreductase activity (Figure 2). However, the literature on the pathophysiologic roles of STEAP1–4 in PCa remains limited. Further studies are needed to better understand these pathophysiologic mechanisms.

## 4. STEAP1–4 as Biomarkers and Therapeutic Targets for Prostate Cancer

Much remains to investigate regarding the tumorigenic mechanisms of STEAP1–4. Nevertheless, that STEAP1, STEAP2, and STEAP4 are upregulated in PCa provides enough impetus to investigate them as possible biomarkers and therapeutic targets for PCa. In this section, we describe efforts to translate STEAP1–4 into the clinic.

### 4.1. STEAP1, STEAP2, and STEAP4 as Biomarkers for Prostate Cancer

Serum PSA is currently the gold-standard biomarker for PCa screening, detection, and prognostication. However, recent PCa screening trials highlighted the limitations of using PSA—namely, that PSA screening does not significantly reduce mortality in PCa patients and is associated with a high risk of overdiagnosis [46,47]. This led the United States Preventive Services Task Force to recommend against routine PSA-based PCa screening, particularly in patients who are 70 or older [48,49]. Since PSA screening does sometimes detect PCa early, the USPSTF recommendation is somewhat controversial. This warrants finding PCa biomarkers with better specificity.

STEAP1, STEAP2, and STEAP4 have been evaluated as possible diagnostic and prognostic biomarkers in glioblastoma, breast cancer, Ewing sarcoma, lung cancer, and PCa [50,51,52,53,54]. While the literature disagrees about the statistical significance of using STEAP1, STEAP2, and STEAP4 for PCa screening, diagnosis, and prognosis, the evidence suggests some degree of clinical utility.

STEAP1 is a promising diagnostic and prognostic biomarker. One method of using STEAP1 as a biomarker is detecting the number of circulating STEAP1-positive extracellular vesicles (EVs). In two studies, PCa patients were found to have significantly elevated STEAP1 EV levels when compared to healthy males [55,56]. Khanna et al. concluded that EV-based liquid biopsy may be a useful diagnostic strategy in PCa, but they found no association between total STEAP1 EV levels and disease recurrence or overall survival [55]. Currently, no EV-based diagnostics for PCa exist. Exploring STEAP1 as a prognostic biomarker, Ihlaseh-Catalano et al. investigated whether STEAP1 overexpression was associated with higher Gleason grades, seminal vesicle invasion, shorter biochemical recurrence-free survival, and higher mortality [57]. Although they found associations between STEAP1 and these prognostic measures, the only significant association was between STEAP1 overexpression and biochemical recurrence. Meanwhile, Gomes et al. showed that STEAP1 is useful for distinguishing malignant PCa from benign prostatic hyperplasia [58]. However, they acknowledged that STEAP1 lacks specificity in distinguishing prostatic intraepithelial neoplasia from PCa.

STEAP1-specific antibodies are being tested as diagnostic imaging agents. A notable example is ^89^Zr-DFO-MSTP2109A, which uses an STEAP1-specific humanized immunoglobulin G1 (MSTP2109A) to target ^89^Zr-desferrioxamine (DFO), a positron emission tomography radionuclide, to mCRPC [59,60]. Ongoing phase I/II clinical trials show that ^89^Zr-DFO-MSTP2109A is well tolerated and demonstrates excellent uptake in bone and soft-tissue mCRPC sites (ClinicalTrials.gov identifier NCT01774071).

STEAP2 and STEAP4 are interesting biomarker candidates because, in the prostate, they are overexpressed in malignant tissue. Burnell et al. showed that STEAP2 overexpression was significantly correlated with the Gleason score, a measure of prognosis [54]. STEAP2 is particularly promising because, in a series of stratified analyses with possible confounders (age, PSA), STEAP2 was found not to significantly correlate with age or PSA. However, a previous study found no significant association between STEAP2 and the Gleason score [33]. Burnell et al. also showed that STEAP4 may be useful in predicting PCa relapse [54]. The number of relapsing patients in the study, though, was small.

Ultimately, further investigation is needed to determine whether STEAP1, STEAP2, and STEAP4 are useful diagnostic and prognostic biomarkers for PCa.

### 4.2. STEAP1 as a Therapeutic Target for Prostate Cancer

PCa-specific antigens that are therapeutic targets include prostate stem cell antigen (PSCA), prostate-specific membrane antigen (PSMA), and STEAP1. This section will only discuss STEAP1. Given its localization to surface cell–cell junctions, particularly that of the prostatic secretory epithelium, STEAP1 is a promising target for T-cell and antibody-based immunotherapy [61,62]. Because STEAP1 is overexpressed in malignant prostate tissue but expressed at low levels in normal prostate tissue, STEAP1 is an ideal therapeutic target (Table 1) [59,60,63,64,65,66,67,68,69,70,71,72,73].

STEAP1 is immunogenic; STEAP1-derived peptides expressed on MHC class I molecules induce cytotoxic CD8+ T lymphocyte (CTL) activity in PCa ex vivo [74,75,76]. These studies found that several epitopes of STEAP1 induce CTL activity, e.g., STEAP_86–94_, STEAP_262–270_, and STEAP_292–300_, making them attractive targets for cancer vaccines. In cell-line and mouse models, vaccination with these epitopes conferred protection against new tumor growth; vaccination against some of these epitopes also attenuated the growth of well-established tumors. Since STEAP1 is a self-antigen, one concern about these vaccines is that they may trigger autoimmunity. Recognizing this possibility, Luz Garcia-Hernandez et al. tested whether STEAP1 vaccines induce autoimmunity in mice [76]. Under physiologic conditions, autoantibodies against STEAP1 were generated at a low but detectable quantity. Notably, though, no pathologic autoimmunity was observed. This may be due to the low basal levels of STEAP1 expression in normal tissues.

Commercially developed PCa vaccines include CureVac’s CV9104, a self-adjuvanted full-length mRNA vaccine that targets the PCa antigens PSA, PSCA, PMSA, STEAP1, prostatic acid phosphatase (PAP), and mucin 1 [77]. Although CV9104 was well tolerated and triggered a robust immune response, its clinical trial (ClinicalTrials.gov identifier NCT01817738) was terminated because CV9104 did not improve overall survival [63]. As of this writing, no active clinical trials are underway for STEAP1 vaccines.

DSTP3086S (vandortuzumab vedotin) is an antibody–drug conjugate (ADC) that, like ^89^Zr-DFO-MSTP2109A, contains MSTP2109A [70]. In DSTP3086S, MSTP2109A is conjugated via a protease–labile linker to the potent antimitotic agent monomethyl auristatin E (MMAE). After antigen-specific binding of DSTP3086S to STEAP1-overexpressing cells, DSTP3086S is endocytosed. MMAE is then released intracellularly, leading to inhibition of cell-cycle progression. DSTP3086S was investigated in a phase I, multicenter, open-label, dose-escalation study (ClinicalTrials.gov identifier NCT01283373). Compared to traditional chemotherapies for mCRPC, DSTP3086S exhibited a relatively mild adverse effect (AE) profile; the most common AEs were fatigue, peripheral neuropathy, and gastrointestinal symptoms. Notably, these were associated with the drug MMAE, not the STEAP1-specific antibody MSTP2109A. Danila et al. noted that, while DSTP3086S requires optimization for further clinical development, STEAP1-targeting ADCs offer promise.

AMG 509 is an Xmab^®^2+1 T-cell engager that contains two identical anti-STEAP1 Fab domains, an anti-CD3 scFv domain, and an Fc domain that prolongs serum half-life [78]. Announcing its development, Li et al. described AMG 509 as having potential treatment utility for mCRPC and Ewing sarcoma. AMG 509 induces CTL-mediated cytotoxicity of STEAP1-positive cancer cells, with a median EC_50_ of 37 pM across 19 STEAP1-expressing cancer cell lines. Notably, the presence of two identical anti-STEAP1 Fab domains (versus one) increased CTL activity against cancer cells and decreased off-target activity against normal cells. The bispecificity of AMG 509 ensures that AMG 509 largely binds cells that overexpress STEAP1, resulting in more robust anti-cancer activity and fewer AEs. A phase I, multicenter, open-label study for assessing the safety, tolerability, pharmacokinetics, and efficacy of AMG 509 is currently recruiting mCRPC patients (ClinicalTrials.gov identifier NCT04221542) [71]. As of this writing, AMG 509 is the only STEAP-related therapeutic that is actively recruiting for clinical trials.

### 4.3. Other Considerations and Perspectives on the Role of STEAP1–4 in Prostate Cancer

Other anti-STEAP1 therapeutics are under investigation for renal cell carcinoma, urothelial carcinoma, and Ewing sarcoma [72,79]. Lin et al. described BC261, a rehumanized STEAP1-IgG that is bispecific for STEAP1 and CD3 [72]. Its design is like that of AMG 509; BC261 consists of two identical anti-STEAP1 Fab domains and two identical anti-CD3 scFv domains. Preliminary results in the Ewing sarcoma, PCa, and canine osteosarcoma cell lines demonstrated significant elevation of T-cell infiltration and tumor ablation. There is, thus, impetus to investigate other anti-STEAP antibody-based agents for therapeutic purposes in cancers other than PCa. Excluding AMG 509 as a possible therapeutic for Ewing sarcoma, these studies have not yet led to clinical trials.

Although STEAP1 is the primary target of current clinical investigation, other STEAP proteins are also the subjects of translational studies. For example, Machlenkin et al. identify STEAP3-derived epitopes as good vaccine candidates for immunotherapy of PCa. In vitro cytotoxicity assays demonstrated that these epitopes induced strong CTL-mediated antitumor activity [80]. Other approaches for targeting STEAP in PCa include fusion protein vaccines, RNA–lipoplex vaccines, recombinant viral vaccines, DNA vaccines, and CAR-T cells [64,65,66,67,73]. These studies are promising, and there is much work to do before they can be translated into PCa therapeutics. The literature on targeting STEAP2-4 is limited, but the approaches used to study STEAP1 can also be used to study STEAP2-4.

## 5. Conclusions

Given the dynamic physiologic and pathologic mechanisms associated with STEAP1–4, they may have potential as therapeutic targets. However, further investigation is needed on many fronts. The remaining questions to answer include: STEAP1–4 expression patterns in different grades of PCa, the role of AR in regulating STEAP1–4 expression, other mechanisms by which STEAP1–4 promote PCa pathogenesis, and the roles of STEAP1–4 in the PCa tumor microenvironment. Answering these questions will allow us to better understand the implications and utility of STEAP1–4 in the clinic.

## Figures and Tables

**Figure 1 cancers-14-04034-f001:**
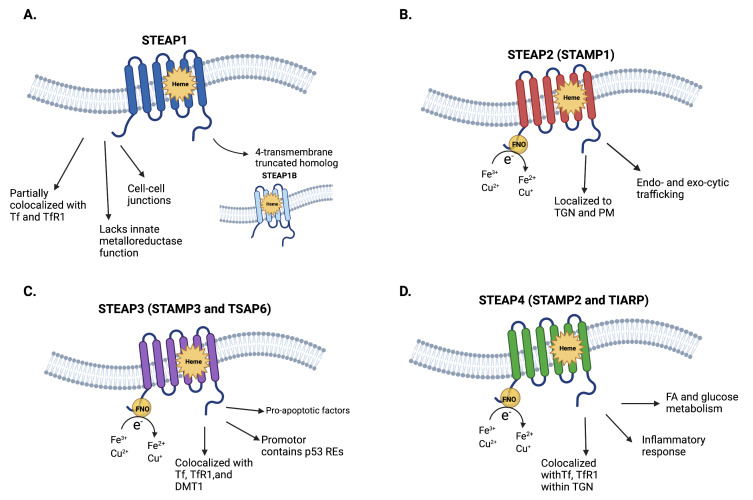
Structural and physiologic properties of STEAP1–4. (**A**) STEAP1 and its truncated homolog, STEAP1B (only STEAP1B1 is displayed). The C-terminal domain of STEAP1 and STEAP1B is homologous with *Saccharomyces cerevisiae* ferric reductase (FRE) and contains a *b*-type heme cofactor. (**B**) STEAP2, (**C**) STEAP3, and (**D**) STEAP4. Like STEAP1 and STEAP1B, STEAP2-4 have C-terminal domains that are homologous with FRE. Unlike STEAP1 and STEAP1B, STEAP2-4 also have N-terminal domains that are homologous with prokaryotic F_420_:NADP^+^ oxidoreductase (represented by a yellow circle, FNO). TGN, *trans*-Golgi network. PM, plasma membrane. Tf, transferrin. TfR1, transferrin receptor 1. DMT1, divalent metal transporter 1. REs, response elements. FA, fatty acid. Fe, iron. Cu, copper.

**Figure 2 cancers-14-04034-f002:**
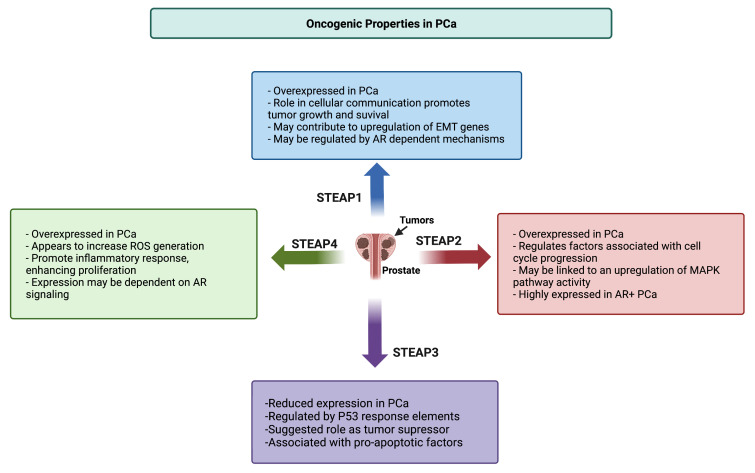
Oncogenic roles of STEAP1–4 in prostate cancer. STEAP1, STEAP2, and STEAP4 are overexpressed in PCa, while STEAP3 expression is downregulated in PCa. Each STEAP protein has its own associated oncogenic mechanisms, many of which remain ill-understood in the literature. PCa, prostate cancer. MAPK, mitogen-activated protein kinase. EMT, epithelial-to-mesenchymal transition. AR, androgen receptor.

**Figure 3 cancers-14-04034-f003:**
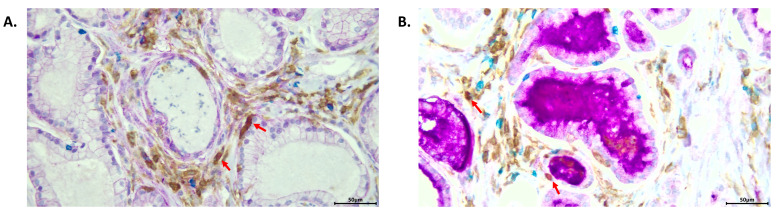
Prostate cancer tumor immune microenvironment. Multiplex immunohistochemical staining of a section of a moderate-grade PCa tumor, magnified 40×. (**A**,**B**) Stains for CD4+ T-cells (brown), CD8+ T-cells (blue), and CD4+/FoxP3+ T-regulatory cells (brown and red, nuclear) are present. Immune cells are seen infiltrating the tumor stroma, with red arrows identifying CD4+/FoxP3+ T-regulatory cells. (**A**) Staining for STEAP1 (purple, membranous) is present. (**B**) Staining for prostate-specific membrane antigen (purple, membranous and cytoplasmic) is present.

**Table 1 cancers-14-04034-t001:** Therapeutic agents that target STEAP1. mRNA, messenger ribonucleic acid. DNA, deoxyribonucleic acid. CAR-T, chimeric antigen receptor T-cell therapy. MSKCC, Memorial Sloan Kettering Cancer Center. FHCC, Fred Hutchinson Cancer Center.

Molecule(s)	Mechanism of Action	Company or Institute/ Trial or Patent	Citation
CV9104	mRNA vaccine	CureVac AG/NCT01817738	[63]
Ag85B-3 × STEAP1_186-193_	Fusion protein vaccine	Tianjin University/N/A	[64]
Unnamed	RNA–lipoplex vaccine	Ahvaz Jundishapur University of Medical Sciences/N/A	[65]
Various, e.g., Ad26.hSTEAP1, MVA.PSMA.hSTEAP1	Recombinant viral vaccine	Janssen/WO2021209897	[66]
PCaA-SEV	DNA vaccine	Inovio/N/A	[67]
Various, e.g., TCT001, TCT002	STEAP1 x CD3, anti-STEAP1 antibodies	Roche/WO2014165818A2	[68]
MSTP2109A	Anti-STEAP1 antibody	Genentech/N/A	[69]
^89^Zr-DFO-MSTP2109A	Antibody–radionuclide conjugate	MSKCC, Genentech/NCT01774071	[59,60]
DSTP3086S (vandortuzumab vedotin)	Antibody–drug conjugate	Genentech/NCT01283373	[70]
AMG 509	Bispecific T-cell-engaging antibody	Amgen/NCT04221542	[71]
BC261	Bispecific T-cell-engaging antibody	MSKCC/N/A	[72]
Unnamed	STEAP1 CAR-T	FHCC/N/A	[73]

## Data Availability

Not applicable.

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
