# Peer review of "STEAP1–4 (Six-Transmembrane Epithelial Antigen of the Prostate 1–4) and Their Clinical Implications for Prostate Cancer"

_cancers, 2022, doi:10.3390/cancers14164034_

Round 1
Reviewer 1 Report
Prostate cancer is the second leading cause of death in men and can develop into metastatic castration-resistant prostate cancer (mCRPC). Since 1994, prostate-specific antigen (PSA) has been the standard in prostate cancer screening. Recently, research has found that this antigen has been associated with false-positive results, leading to unnecessary exposure to radiation and/or castration. As research continues, it has been suggested that a family of metalloproteinases, STEAP1-4 or Six Transmembrane Epithelial Antigen of the Prostate 1-4, should be the biomarker of prostate cancer screening. This enzyme has been seen to participate in the metabolism of iron and copper and its overexpression has been associated with tumorigenesis and metastasis of cancer cells. In this review, the author's research is to detail the pros and cons of each STEAP biomolecule and how they would relate to the development of a new prostate cancer screening. The authors also detail how STEAP1 is the therapeutic target in clinical trials.
The manuscript is well written and well-researched. The authors mention a type of stain used in the immuno-histochemical assessment of prostate cancer by pathologists. To achieve a better sense of what a Gleason score of 3+3 is, an illustration of the Gleason stain ranking would help. This would also provide a better understanding of its importance and the impact that ranking would have on a prostate cancer patient.
It was fascinating to learn about STEAP1-4’s involvement in prostate cancer, but it felt like it was an afterthought rather than the main purpose of this review. The biochemical structures and physiology seem to be the highlight of the manuscript. More mention of prostate cancer’s burden on the population is needed, especially with mCRPC and its interaction with STEAP. This would tie in with the importance of STEAP as a biomarker for prostate cancer screening. Also, the biochemical structures of STEAP1 and its subtypes should be incorporated in the same section. This will provide a more cohesive article. The clinical implication of STEAP as a therapeutic target is well written and researched.
The authors provide a scientifically sound manuscript. The references are up to date with the latest publications, which provide a scientific weight to the manuscript. This review is comprehensive and will add an important aspect to the research of STEAP and its involvement in prostate cancer.
Author Response
- The manuscript is well written and well-researched. The authors mention a type of stain used in the immuno-histochemical assessment of prostate cancer by pathologists. To achieve a better sense of what a Gleason score of 3+3 is, an illustration of the Gleason stain ranking would help. This would also provide a better understanding of its importance and the impact that ranking would have on a prostate cancer patient.
The authors thank the reviewer for their time and favorable assessment of our manuscript. We agree that Gleason scoring should be better explained in the text. As such, we have written an introduction to Gleason scoring and how higher Gleason scores are associated with certain antigen (e.g., STEAP1) expression changes. The main purpose of Figure 3 is to illustrate how highly expressed STEAP1 is on prostate cancer cells. To make this point clearer to readers, we have removed the Gleason score of the tissue sample in Figure 3. Ongoing studies are evaluating the expression of different STEAP proteins in the full range of Gleason score classifications.
- It was fascinating to learn about STEAP1-4’s involvement in prostate cancer, but it felt like it was an afterthought rather than the main purpose of this review. The biochemical structures and physiology seem to be the highlight of the manuscript.
The authors agree that there is a paucity of information regarding STEAP1-4 pathophysiology in prostate cancer in the manuscript. Unfortunately, this paucity reflects, in large part, a lack of literature on the involvement of STEAP1-4 in prostate cancer. One of the goals of our review is to highlight areas that require further research. As such, we have added a sentence to explicitly state the lack of literature on STEAP1-4.
- More mention of prostate cancer’s burden on the population is needed, especially with mCRPC and its interaction with STEAP. This would tie in with the importance of STEAP as a biomarker for prostate cancer screening.
We agree that the manuscript should do a better job describing mCRPC and the importance of STEAP as a biomarker for prostate cancer. We have written more about the burden of mCRPC, prostate cancer treatment, and prostate cancer progression to mCRPC in the introduction.
- Also, the biochemical structures of STEAP1 and its subtypes should be incorporated in the same section. This will provide a more cohesive article. The clinical implication of STEAP as a therapeutic target is well written and researched.
We acknowledge the reviewer's point about combining the physiology and pathophysiology sections of the review. We pondered doing so while writing the manuscript; however, we felt it better to separate the two so that different readers could focus on different aspects of STEAP1-4 biology. That is, we wanted to have one section (normal physiology) for readers interested in studying the normal biochemical mechanisms of STEAP1-4 and another section (pathophysiology) for readers interested in translational studies. As such, we hope that the reviewer understands that this separation has a specific purpose for the readers.
Reviewer 2 Report
The review has valuable information for researchers and clinical doctors who are interested in treating prostate cancers and others.
A minor point:
At the first sentence in page 6, " STEAP2-siRNA-transfected cells have an increased proliferative phenotype" should be revised as " STEAP2-siRNA-transfected cells have an decreased proliferative phenotype.
Author Response
- The review has valuable information for researchers and clinical doctors who are interested in treating prostate cancers and others. A minor point: At the first sentence in page 6, "STEAP2-siRNA-transfected cells have an increased proliferative phenotype" should be revised as "STEAP2-siRNA-transfected cells have a decreased proliferative phenotype."
The authors thank the reviewer for their time, favorable assessment, and correction. We agree with it and have edited the manuscript accordingly.